# CGA-MGAN: Metric GAN Based on Convolution-Augmented Gated Attention for Speech Enhancement

**DOI:** 10.3390/e25040628

**Published:** 2023-04-06

**Authors:** Haozhe Chen, Xiaojuan Zhang

**Affiliations:** 1Aerospace Information Research Institute, Chinese Academy of Sciences, Beijing 100190, China; 2Key Laboratory of Electromagnetic Radiation and Sensing Technology, Chinese Academy of Sciences, Beijing 100190, China; 3School of Electronic, Electrical, and Communication Engineering, University of Chinese Academy of Sciences, Beijing 100049, China

**Keywords:** CGA-MGAN, gated attention unit, speech enhancement

## Abstract

In recent years, neural networks based on attention mechanisms have seen increasingly use in speech recognition, separation, and enhancement, as well as other fields. In particular, the convolution-augmented transformer has performed well, as it can combine the advantages of convolution and self-attention. Recently, the gated attention unit (GAU) was proposed. Compared with traditional multi-head self-attention, approaches with GAU are effective and computationally efficient. In this CGA-MGAN: MetricGAN based on Convolution-augmented Gated Attention for Speech Enhancement, we propose a network for speech enhancement called CGA-MGAN, a kind of MetricGAN based on convolution-augmented gated attention. CGA-MGAN captures local and global correlations in speech signals at the same time by fusing convolution and gated attention units. Experiments on Voice Bank + DEMAND show that our proposed CGA-MGAN model achieves excellent performance (3.47 PESQ, 0.96 STOI, and 11.09 dB SSNR) with a relatively small model size (1.14 M).

## 1. Introduction

Speech enhancement (SE) systems are usually used at the frontend of automatic speech recognition processes [1], communication systems [2], and hearing aids [3] to remove noise from speech. Methods based on traditional signal processing, such as subtraction [4], Wiener filtering [5], and minimum mean square estimation [6], are widely used in speech enhancement. Although these methods perform well in handling stationary noises, it is challenging to deal with nonstationary noises. With the development of deep neural networks (DNNs), this model has been used more frequently in speech enhancement in recent years.

The traditional time–frequency domain DNN model reconstructs the speech magnitude spectrum by estimating the mask function [7,8,9] or directly predicting the magnitude spectrum of clean speech [10], ignoring the role of phase information. However, phase information improves speech perception quality under a low signal-to-noise ratio (SNR) [11,12]. In [13], researchers proposed recovering magnitude and phase information simultaneously in the time–frequency (TF) domain by estimating the complex ratio mask (CRM) [14]. However, due to the compensation effect between the magnitude and phase [15], the simultaneous enhancement of magnitude and phase reduces the effect of magnitude estimation [16]. In [17], researchers proposed a decoupling-style phase-aware method. By building a two-path network, the magnitude is estimated to be increased first. Then, the spectrum is refined using residual learning, which can effectively alleviate the problem of the compensation effect.

The self-attention mechanism [18,19] can model the global context, but it is not good at extracting fine-grained local feature patterns. Convolution [20,21,22] is good at capturing local feature information but needs to improve its capture of global information. The convolution-augmented transformer (conformer) [23,24,25] combines convolution and self-attention and models local speech information by inserting deep convolution into the transformer, which can better extract the features of speech signals. However, the computational complexity of the network increases due to the large number of multi-head self-attention (MHSA) structures and feed-forward modules used in multiple stacked conformer blocks.

In speech enhancement, people usually care most about the quality or clarity of speech. Therefore, objective evaluation metrics considering human perception as the cost function can improve speech quality more directly. Nevertheless, standard metrics, such as the perceptual evaluation of speech quality (PESQ), are nondifferentiable. Therefore, they cannot be used directly as cost functions. MetricGAN can mimic evaluation functions such as PESQ by building neural networks. Considering that the objective function based on point distance may not fully reflect the perception difference between noisy and clean speech signals, Ref. [26] introduces MetricGAN [27] to the conformer, which allows the net to use the evaluation metric score learned by the metric discriminator to optimize the generator.

In this paper, we propose a convolution-augmented gated attention MetricGAN called CGA-MGAN, composed of a generator and a discriminator. We use the convolution-augmented gated attention unit (CGAU) to extract speech features in the generator. Compared with the conformer, which used MHSA, the CGAU model we propose allows the network to use weaker single-head self-attention (SHSA). CGAU can capture the local features and global information of speech simultaneously and minimize quality loss with faster speed, lower memory occupation, and better effect. The discriminator can estimate a black-box, nondifferentiable metric to guide the generator in enhancing speech.

Our main contributions can be summarized as the following points:We construct an encoder–decoder structure including gating blocks using the decoupling-style phase-aware method that can collaboratively estimate the magnitude and phase information of clean speech in parallel and avoid the compensation effect between magnitude and phase;We propose a convolution-augmented gated attention unit that can capture time and frequency dependence with lower computational complexity and achieve better results than the conformer;The proposed approach is superior to the previous approaches on the Voice Bank + DEMAND dataset [28], and an ablation experiment has verified our design choice.

The remainder of this paper is organized as follows: Section 2 introduces the related work of speech enhancement. Section 3 analyzes the specific architecture of the CGA-MGAN model we propose. Section 4 introduces the experimental setup, including the dataset used for the experiment, the network’s training setup, and the experimental results’ evaluation indicators. Section 5 analyzes the experimental results, compares them with some existing models, and conducts an ablation experiment. Finally, Section 6 summarizes this work and suggests some future research directions.

## 2. Related Works

This paper focuses on a convolution-augmented gated attention MetricGAN model for speech enhancement. In this section, we briefly introduce MetricGAN, conformers, and their basic working principles. In addition, we review the structure of the standard transformer and briefly introduce the basic principle of the new transformer variant, the gated attention unit (GAU).

### 2.1. MetricGAN

Before introducing MetricGAN, we will first introduce how to use the general GAN network for speech enhancement. GAN can simulate real data distribution by employing an alternative mini-max training scheme between the generator and the discriminator. By using the least-squares GAN method [29] to minimize the following loss function, we can train the generator to map noisy speech, x, to clean speech, y, and to generate enhanced speech.
(1)$$L_{G\left(LSGAN\right)}=E_{x}\left[D\left(G\left(x\right),x\right)-1{)}^{2}\right]$$Here, *G* represents the generator and *D* represents the discriminator. By minimizing the following loss function, we can train *D* to distinguish between clean speech and enhanced speech:(2)$$L_{D\left(LSGAN\right)}=E_{x,y}\left[{\left(D\left(y,x\right)-1\right)}^{2}+{\left(D\left(G\left(x\right),x\right)-0\right)}^{2}\right]$$Here, *E* represents the expectation, 1 represents the clean speech, and 0 represents the enhanced speech. G(x) represents the enhanced speech, and D(·) represents the output result of the discriminator.

MetricGAN consists of a generator and a discriminator, the same as the general GAN network. The generator enhances speech. The discriminator treats the objective evaluation function as a black box and trains the surrogate evaluation function. During training, the discriminator and the generator are updated alternately to guide the generator to generate higher-quality speech.

Unlike the general GAN network, MetricGAN introduces a function (Q(I)) to represent the normalized metric to be simulated. The loss functions of MetricGAN are shown in the following formulas:(3)$$L_{G\left(MetricGAN\right)}=E_{x}\left[{\left(D\left(G\left(x\right),y\right)-s\right)}^{2}\right]$$
(4)$$L_{D\left(MetricGAN\right)}=E_{x,y}\left[{\left(D\left(y,y\right)-Q\left(y,y\right)\right)}^{2}+{\left(D\left(G\left(x\right),y\right)-Q\left(G\left(x\right),y\right)\right)}^{2}\right]$$

Formula (3) is the loss function of the generator network, where s represents the expected distribution score. When s = 1, the generator will generate enhanced speech that is close to clean speech.

Formula (4) is the loss function of the discriminator network, where Q(I) represents the function of the target evaluation metric normalized between 0 and 1, and I represents the speech pair to be evaluated. When the inputs are both clean voices, I=(y,y); when the inputs are an enhanced voice and a corresponding clean voice, I=(G(x),y).

The training process for MetricGAN can be condensed into the following schema:

Input noisy speech, x, into the generator to generate enhanced speech, G(x);Input a clean–clean speech pair, (y,y), into the discriminator to calculate the output, D(y,y), and calculate Q(y,y) through the objective evaluation function;Input an enhanced–clean speech pair, (G(x),y), into the discriminator to calculate the output, D(G(x),y), and calculate Q(G(x),y) through the objective evaluation function;Calculate the loss function of the generator and the discriminator and update the weights of both networks.

### 2.2. Gated Attention Unit

Recently, Hua W. [30] proposed a new transformer variant called GAU. Compared with the standard transformer, it has faster speed, lower memory occupation, and a better effect.

The standard transformer comprises, alternately, an attention block and a feed-forward network (FFN) layer, which consists of two multi-layer perceptron (MLP) layers. The attention block uses MHSA, as shown in Figure 1a. Unlike the standard transformer, GAU has only one layer, which makes networks stacked with GAU modules simpler and easier to understand. GAU creatively uses the gated linear unit (GLU) instead of the FFN layer. The structure of the GLU is shown in Figure 1b. The powerful performance of GLU allows GAU to weaken its dependence on attention. GAU can use SHSA instead of MHSA, achieving the same or even better effects compared with the standard transformer [30]. It not only improves the computing speed but also reduces memory occupation. The structure of GAU is shown in Figure 2.

### 2.3. Conformer

The conformer was first used in speech recognition and can also be used for speech enhancement. Since a pronunciation unit is composed of multiple adjacent speech frames, the convolution mechanism can better extract fine-grained local feature patterns, such as pronunciation unit boundaries. The conformer combines the convolution and self-attention modules and gives full play to their advantages. The main structure of the conformer is shown in Figure 3. The conformer block consists of four parts: the first feed-forward module, an MHSA module, a convolution module, and the second feed-forward module. The detailed structure of the convolution block is shown in Figure 4, and the detailed structure of the feed-forward module is shown in Figure 5. Inspired by Macaron Net [31], the conformer adopts a makaron-style structure. The convolution module and the MHSA module are placed between two feed-forward modules. By stacking the conformer blocks, speech features are extracted step by step to achieve speech recognition or speech enhancement.

### 2.4. Limitations and Our Approach

MetricGAN is a great contribution to the application of GAN to speech enhancement. GAN simulates evaluation metrics that were originally nondifferentiable so that it can be used as a loss function. However, the performance of the MetricGAN generator limits its speech enhancement effect. In our CGA-MGAN model, we use the idea of MetricGAN to build a discriminator and a generator with an encoder–decoder structure, including gating blocks using the decoupling-style phase-aware method, which can greatly improve the network’s speech enhancement performance.

In addition, although GAU has been applied to natural speech processing, previous research has yet to involve the field of speech enhancement. This paper is the first application of GAU to speech enhancement. Compared with makaron-style structures used in conformers, the CGAU we propose uses GLU to replace two feed-forward modules in the conformer, replaces MHSA with SHSA, and perfectly integrates the convolution module and GAU, significantly reducing the computational complexity of the network. A convolution-augmented GAU constructed this way can extract global and local features simultaneously to obtain better speech quality.

## 3. Methodology

In this section, we introduce the composition of the CGA-MGAN model, including the encoder–decoder structure of the generator, the structure of the two-stage CGA block, and the structure of the metric discriminator. Finally, we introduce the loss function of the generator and the metric discriminator.

The architecture of the generator is shown in Figure 6. The generator consists of an encoder–decoder structure, including gating blocks using the decoupling-style phase-aware method and four two-stage CGA blocks. First, it takes a discrete noisy signal, y∈ℝB×N×1, with N samples as the input. Then, we convert the input signal to Yo∈ℝB×T×F×1 in a time-frequency representation domain using short-time Fourier transform (STFT), where T represents the number of frames and F represents the number of frequency bins of the complex spectrogram. After that, a power law compression with a compression exponent of 0.3 is applied to the spectrum:(5)Y=|Yo|cejYp=YmejYp=Yr+jYi
where *c* is the compression exponent. Then, the magnitude, Ym, real component, Yr, and imaginary component, Yi, of the spectrum are concatenated as Yin=[Ym;Yr;Yi]∈ℝB×T×F×3 as the input of the encoder.

### 3.1. Encoder and Decoder

The encoder of the generator consists of five encoder blocks with concatenation operations. The last encoder block halves the frequency dimension to reduce complexity. Each encoder block consists of a Conv2D (two-dimensional convolution) layer, an instance normalization layer, and a parameter ReLU (PReLU) activation layer. The output feature of the encoder is Yenc∈ℝB×T×F′×C, where F′=F/2, C=64.

The decoder of the generator consists of three decoder blocks, including a magnitude mask estimation decoder block and two complex spectrum estimation decoder blocks, which output the multiplicative mask of the magnitude, ${{\hat{X}}^{\prime }}_{m}$; the real component, ${{\hat{X}}^{\prime }}_{r}$; and the imaginary component, ${{\hat{X}}^{\prime }}_{i}$, of the spectrum in parallel. Each decoder block consists of five gated blocks and a Conv2D layer. The first gated block samples the frequency dimension up to F. The gated block consists of a Conv2D Transpose layer and two Conv2D blocks. The structure of the Conv2D block in the decoder block is the same as that of the encoder block. The gated block learns features from the encoder and suppresses its unwanted parts, which is shown in Figure 7. After five decoder blocks, the Conv2D layer compresses the number of channels to obtain ${{\hat{X}}^{\prime }}_{m}$, ${{\hat{X}}^{\prime }}_{r}$, ${{\hat{X}}^{\prime }}_{i}$. Then, we multiply the ${{\hat{X}}^{\prime }}_{m}$ and magnitude $Y_{m}$ of noisy speech to obtain the preliminary estimated spectrum.

As a supplement to spectrum estimation, the preliminary estimated spectrum is coupled with the noisy speech phase, $Y_{p}$, to obtain a roughly denoised complex spectrum diagram. Then, it is added element wise with the output (${{\hat{X}}^{\prime }}_{r}$,${{\hat{X}}^{\prime }}_{i}$) of the complex spectrum estimation decoder block to obtain the final complex spectrum diagram:(6)$${\hat{X}}_{r}={{\hat{X}}^{\prime }}_{m}Y_{m}\cos Y_{P}+{{\hat{X}}^{\prime }}_{r}$$
(7)$${\hat{X}}_{i}={{\hat{X}}^{\prime }}_{m}Y_{m}\sin Y_{P}+{{\hat{X}}^{\prime }}_{i}$$
(8)$${\hat{X}}_{m}=\sqrt{{\hat{X}}_{r}{}^{2}+{\hat{X}}_{i}{}^{2}}$$
where ${\hat{X}}_{m}$, ${\hat{X}}_{r}$ and ${\hat{X}}_{i}$ represent the magnitude, the real component of spectrum and the imaginary component of spectrum of the enhanced speech.

### 3.2. Two-Stage CGA Block

The two-stage CGA block consists of two cascaded CGAUs, namely, the time convolution-augmented gated attention unit (CGAU-T) and the frequency convolution-augmented gated attention unit (CGAU-F), which is shown in Figure 8. First, the input feature map, D∈ℝB×T×F′×C, is reshaped to DT∈ℝBF′×T×C and input into CGAU-T to capture the time dependence. Then, the DoT and DT are element-wise added and reshaped to DF∈ℝBT×F′×C and input into CGAU-F to capture the frequency dependence. Finally, output DoF and input DF are element-wise added and reshaped to the final output, Do∈ℝB×T×F′×C.

The CGAU-T and CGAU-F blocks have the same structure and different shaping operations. The structure of CGAU is shown in Figure 9 and is composed of a convolution block and a GAU. The input is connected to the output by a residual connection. We use the same structure as the convolution block in the conformer. The convolution block starts with a layer normalization. After that, the feature map is fed into a gating mechanism composed of a point-wise convolution, followed by GLU. Then, the output of the GLU is fed into a depth-wise convolution layer and activated by the swish function. Finally, a point-wise convolution layer restores the channel number.

Taking CGAU-T as an example, input DT is divided into two feeds, one of which is fed into the convolution Block. The query, key, and value are all replicas of the convolution block output, DcoT∈ℝBF′×T×C. Then, the scaled dot-product attention is applied to the query, key, and value afterward as
(9)Z=∅z(DcoTWz)
(10)V=∅v(DcoTWv)
(11)A=softmax(Q(Z)K(Z)⊺+bd)V
where ∅ represents the swish activation function, and Wz and Wv represent the learnable parameter matrixes of the linear layers. Z is a shared representation; Q and K are simple affine transformations that apply per-dim scalars and offsets to Z to obtain the query and the key. V represents values in the self-attention mechanism. b represents the rotation position coding [32]. d represents the dimension.

The other feed of input $D^{T}$ passes through the linear layer and is activated by swish to obtain U. Finally, the Hadamard product of U and A is calculated and input into the linear layer to obtain output $D_{o}^{T}$ so that the convolution-augmented attention information is introduced to the gated linear unit. The calculation formula is as follows:(12)$$U=\emptyset_{u}\left(D^{T}W_{u}\right)$$
(13)$$D_{o}^{T}=\left(U\circ A\right)W_{o}$$
where ∘ represents Hadamard product. In addition, our proposed CGAU uses softmax as the activation function instead of the ${\mathrm{ReLU}}^{2}$ used in GAU.

### 3.3. Metric Discriminator

The metric discriminator can mimic the metric score, which is nondifferentiable, so that it can be used as the loss function. In this paper, we use the normalized PESQ as the metric score. As shown in Figure 10, the metric discriminator consists of four Conv2D layers. The channels are 32, 64, 128, and 256. After four Conv2D layers, there is a global average pooling to handle the variable-length input. Finally, there are two linear layers and one sigmoid activation. When training the discriminator, we take both inputs as clean magnitudes to estimate the maximum metric score. Then, we take the inputs as the clean magnitudes and the enhanced magnitudes to estimate the metric score of the enhanced speech to approach their corresponding PESQ label. In addition, the trained generator can render enhanced speech approaching a normalized PESQ label of one.

### 3.4. Loss Function

The loss function of the generator includes three terms:(14)$$L_{G}=\alpha L_{TF}+\beta L_{GAN}+\gamma L_{time}$$
where α, β, and γ are the weighting coefficients of the three loss terms in the total loss. Here, we take α as 1, β as 0.05, and γ as 0.2. $L_{TF}$ represents the combination of magnitude loss, $L_{Mag}$, and phase-aware loss, $L_{RI}$:(15)$$L_{Mag}=E_{X_{m},{\hat{X}}_{m}}\left[\|X_{m}-{\hat{X}}_{m}{\|}^{2}\right]$$
(16)$$L_{RI}=E_{X_{r},{\hat{X}}_{r}}\left[\|X_{r}-{\hat{X}}_{r}{\|}^{2}\right]+E_{X_{i},{\hat{X}}_{i}}\left[\|X_{i}-{\hat{X}}_{i}{\|}^{2}\right]$$
(17)$$L_{TF}=mL_{Mag}+\left(1-m\right)L_{RI}$$
where m represents the chosen weight, and we take m=0.7. $L_{GAN}$ represents the adversarial loss. The expression of $L_{GAN}$ is
(18)$$L_{GAN}=E_{X_{m},{\hat{X}}_{m}}\left[\|D\left(X_{m}-{\hat{X}}_{m}\right)-1{\|}^{2}\right]$$
where D represents the discriminator. Correspondingly, the expression of the adversarial loss in the discriminator is
(19)$$L_{D}=E_{X_{m}}\left[\|D\left(X_{m}-X_{m}\right)-1{\|}^{2}\right]+E_{X_{m},{\hat{X}}_{m}}\left[\|D\left(X_{m}-{\hat{X}}_{m}\right)-Q_{PESQ}{\|}^{2}\right]$$
where $Q_{PESQ}$ is the normalized PESQ score, and the value range is [0, 1]. In addition, some studies show that adding time loss, $L_{time}$, can improve the enhancement of speech [20]. The expression of $L_{time}$ is
(20)$$L_{time}=E_{x,\hat{x}}\left[\|x-{\hat{x}\|}_{1}\right]$$

## 4. Experiments

In this section, we first introduce the composition of the Voice Bank + DEMAND dataset for network training; then, we introduce the network training settings and six evaluation indicators for voice enhancement.

### 4.1. Datasets and Settings

The publicly available Voice Bank + DEMAND dataset was chosen to test our model. The speech database was obtained from the CSTR VCTK Corpus. The background noise database was obtained from the DEMAND database. The training set includes 11,572 sentences provided by 28 speakers, and the test set includes 824 sentences provided by 2 unseen speakers. We use eight natural and two artificial background noise processes to generate the training set under different SNR levels, ranging from 0 to 15 dB with an interval of 5 dB. We use five unseen background noise processes to generate the test set under different SNR levels, ranging from 2.5 to 17.5 dB with an interval of 5 dB.

All sentences are resampled to 16 kHz. For the training set, we slice the sentences into 2 s units, but there is no slicing in the test set. We use a Hamming window of length 25 ms and a hop length of 6.25 ms. Since we apply a power law compression with a compression coefficient of 0.3 to the spectrum [33] after STFT, it is reversed on the final estimated complex spectrum. Finally, we apply the inverse STFT to recover the time-domain signal. In addition, we use the AdamW optimizer to train both generator and discriminator for 100 epochs. The batch size is set to four. The learning rate of the discriminator is set to 0.001, which is twice that of the generator. After every 30 epochs, both learning rates will be halved.

### 4.2. Evaluation Indicators

We use six objective measures to evaluate the quality of the enhanced speech. For all metrics, higher scores indicate better performance.

PESQ [34]: Ranges from −0.5 to 4.5;

CSIG [35]: Mean opinion score (MOS) prediction of the signal distortion; ranges from 1 to 5;

CBAK [35]: MOS prediction of the background noise intrusiveness; ranges from 1 to 5;

COVL [35]: MOS prediction of the overall effect; ranges from 1 to 5;

SSNR: The segmental signal-to-noise ratio; ranges from 0 to ∞;

STOI [36]: The short-time objective intelligibility; ranges from 0 to 1.

## 5. Results and Discussion

In this section, we first conduct a comparison with baselines to verify the performance of the proposed model. Then, the effectiveness of CGA-MGAN is verified using ablation experiments. The experimental results are discussed and explained.

### 5.1. Baselines and Results Analysis

The CGA-MGAN we propose can be compared with some existing models objectively. These models include classical models, such as SEGAN [37], DCCRN [13], and Conv-TasNet [20], and state-of-the-art (SOTA) baselines. SEGAN is the earliest application of GAN in speech enhancement, and its network training is conducted by directly inputting time domain waveforms. DCCRN builds a complex convolution recursive network to simultaneously recover the magnitude and phase information in the TF domain by estimating the CRM. Conv-TasNet is a fully convolutional end-to-end time-domain speech separation network that can also be used for speech enhancement.

For methods based on the generation model, we choose three baselines, including TDCGAN [38], MetricGAN+ [39], UNIVERSE [40], and CDiffuSE [41]. TDCGAN first introduced dilated convolution and deep separable convolution to the GAN network and can build a time-domain speech enhancement system. It dramatically reduces network parameters and obtains a better speech enhancement effect than SEGAN. MetricGAN+ is an improved MetricGAN network for speech enhancement. It not only uses enhanced and clean speech to train the discriminator, but also uses noisy speech to minimize the distance between the discriminator and target objective metrics. Moreover, MetricGAN+’s generator uses the learnable sigmoid function for mask estimation, which improves the generator’s speech enhancement ability. UNIVERSE builds a generative score-based diffusion model and a multi-resolution conditioning network so that mixture density networks can be enhanced and achieve good speech enhancement results. CDiffuSE learns the characteristics of noise signals and incorporates them into diffusion and reverse processes, making the model highly robust against changes in noise characteristics.

For methods based on the improved transformer, we choose four baselines, including SE-Conformer [25], DB-AIAT [17], DPT-FSNet [42], and DBT-Net [43]. SE-Conformer first introduced the convolution-augmented transformer to the field of speech enhancement. The proposed speech enhancement architecture can focus on the whole sequence through MHSA and convolutional neural networks to capture short-term and long-term time series information. DB-AIAT constructs a dual-branch network through the decoupling-style phase-aware method. It first estimates the magnitude spectrum roughly, and then the spectral details that the magnitude-marking branch missed are compensated. DPT-FSNET integrates subband band and complete band information and proposes a transformer-based dual-branch frequency domain speech enhancement network. DBT-Net proposes a dual-branch network to estimate magnitude and phase information. Interaction modules are introduced to obtain features learned from one branch to facilitate the information flow between branches to curb the undesired parts.

As shown in Table 1, compared with the improvement work of MetricGAN (MetricGAN+), there is a 0.32 improvement in the PESQ score. In addition, compared with the advanced generation model currently used in speech enhancement, CGA-MGAN achieves better performance. Finally, compared with recent methods based on improved transformers, such as DPT-FSNet, our method is also better in almost all evaluation scores. At the same time, the model size is relatively small (1.14 M).

### 5.2. Ablation Study

To investigate the contribution of the different CGA-MGAN components proposed to enhance performance, we have conducted an ablation study. Several variants of the CGA-MGAN model are compared in Table 2: (i) removing the convolution block in CGAU; (that is, using GAU to replace CGAU (w/o Conv. Block)); (ii) using conformer to replace CGAU in the two-stage block (using a conformer); (iii) removing gating blocks in the decoder (w/o gating decoders); (iv) removing the phase compensation mechanism and only improving the speech in the magnitude spectrum (Mag-only); (v) removing the discriminator (w/o discriminator).

We set all variants using the same configuration as CGA-MGAN. As shown in Table 2, all these variants underperform the proposed CGA-MGAN. Comparing CGA-MGAN with (i), a decrease of 0.1 in PESQ can be observed because GAU cannot extract the short-term features of speech very well after the convolution block is removed. Comparing CGA-MGAN with (ii), a decrease of 0.14 in PESQ can be observed. This proves that the CGAU model we propose is superior to traditional convolution-augmented transformers for speech enhancement. Comparing CGA-MGAN with (iii) and (iv), we find that gating blocks and the phase-aware method are essential to CGA-MGAN. Because the gating block can retain important features in the encoder and suppresses irrelevant features, it can also control the information flow of the network and simulate complex interactions, which is quite effective for improving the network’s speech enhancement ability [22]. In addition, the decoupling-style phase-aware method can process the coarse-grained regions and fine-grained regions of the spectrum in parallel so that lost spectrum details can be compensated for, and it can avoid the compensation effect between magnitude and phase [17]. Finally, comparing CGA-MGAN with (v), we can observe that removing the discriminator has a negative impact on all given scores, which proves the advantages of using MetricGAN.

In addition, we can focus on CGA-MGAN and (ii) and use real-time factors (RTFs) to compare their computational complexity. RTFs can be measured by taking an average of five runs on an Intel Xeon Silver 4210 CPU. The RTF of CGA-MGAN is 0.017, while the RTF of (ii) is 0.025. This proves that the CGAU we propose has lower computational complexity than traditional convolution-augmented transformers.

## 6. Conclusions

In this work, we propose CGA-MGAN for speech enhancement, which can better combine the advantages of convolution and self-attention. Our approach combines CGAU, which can capture time and frequency dependencies with lower computational complexity and achieve better results, together with an encoder–decoder structure, including gating blocks using the decoupling-style phase-aware method, which can collaboratively estimate the magnitude and phase information of clean speech and avoid the compensation effect. Experiments on Voice Bank + DEMAND show that the CGA-MGAN model we propose performs excellently at a relatively small model size (1.14 M). In the future, we will further study the application of CGAU to other speech tasks, such as separation and dereverberation.

## Figures and Tables

**Figure 1 entropy-25-00628-f001:**
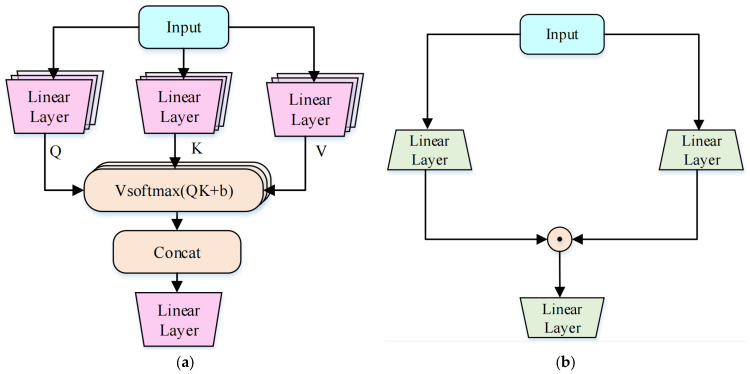
(**a**) Muti-head self-attention; (**b**) gated linear unit.

**Figure 2 entropy-25-00628-f002:**
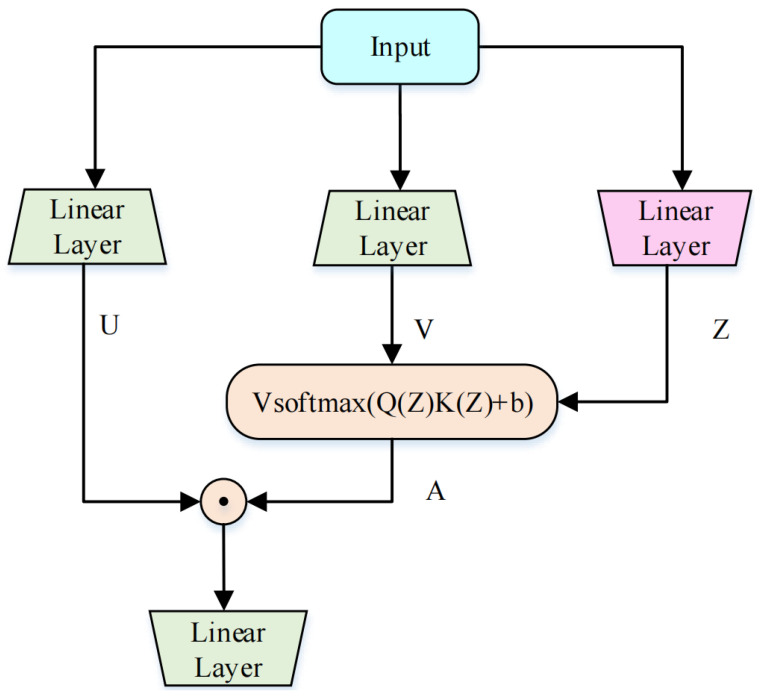
Gated attention unit.

**Figure 3 entropy-25-00628-f003:**
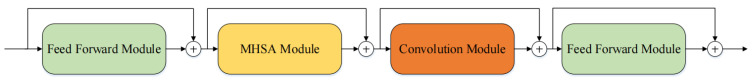
Conformer architecture.

**Figure 4 entropy-25-00628-f004:**
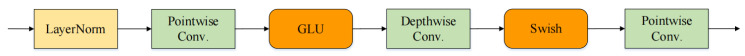
Detailed structure of the convolution block.

**Figure 5 entropy-25-00628-f005:**
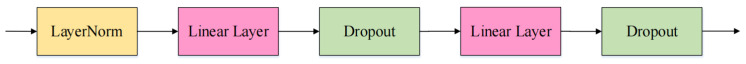
Detailed structure of the feed-forward module.

**Figure 6 entropy-25-00628-f006:**
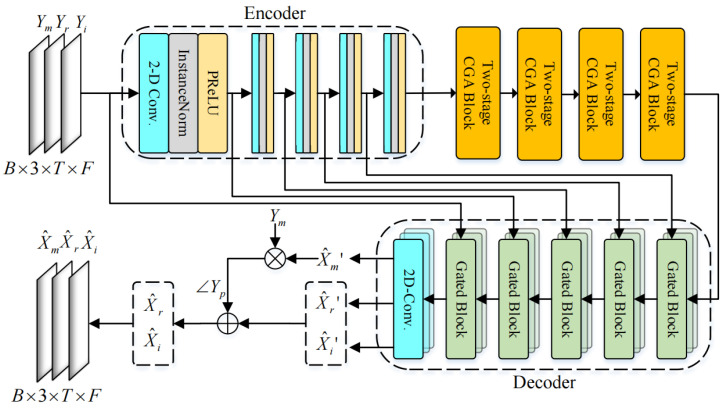
Encoder–decoder generator architecture.

**Figure 7 entropy-25-00628-f007:**
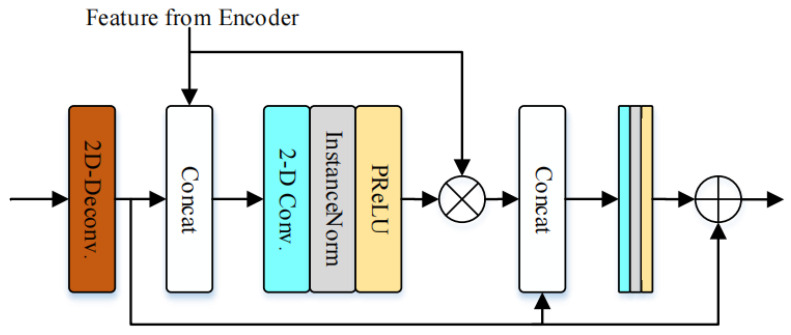
Detailed structure of the gated block inside the decoder.

**Figure 8 entropy-25-00628-f008:**
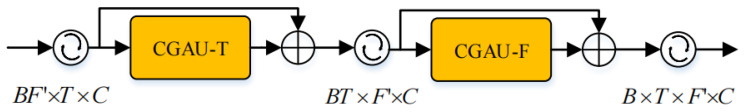
Two-stage CGA block architecture.

**Figure 9 entropy-25-00628-f009:**
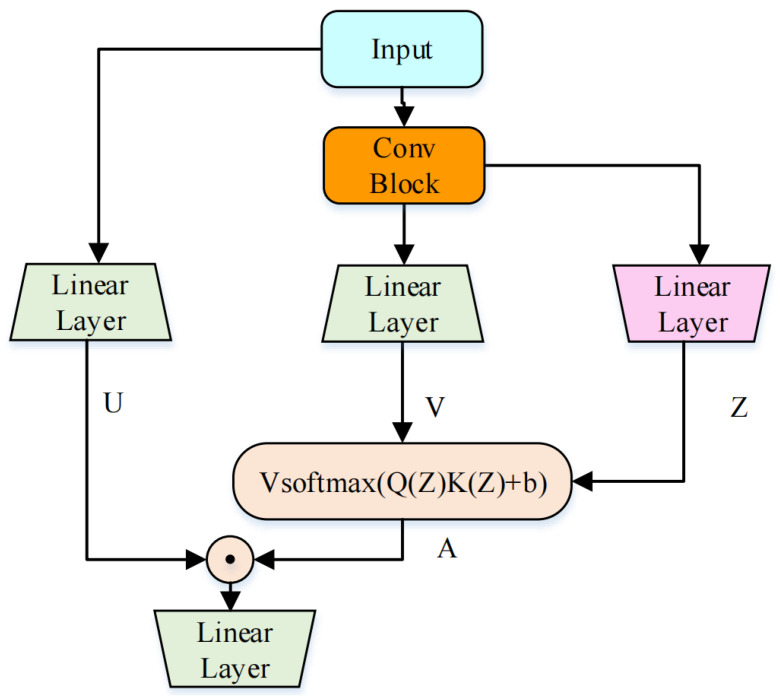
Our proposed convolution-augmented gated attention unit.

**Figure 10 entropy-25-00628-f010:**
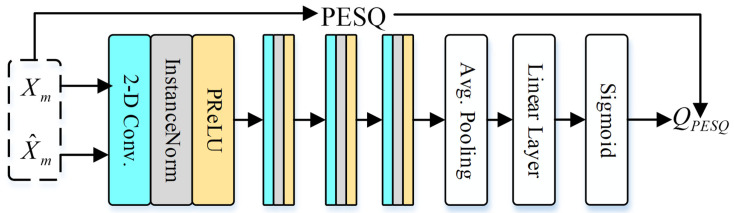
Detailed structure of the metric discriminator.

**Table 1 entropy-25-00628-t001:** Performance comparison of Voice Bank + DEMAND dataset.

Method	Size(M)	PESQ	CSIG	CBAK	COVL	SSNR	STOI
Noisy	- *	1.97	3.35	2.44	2.63	1.68	0.91
SEGAN	97.47	2.16	3.48	2.94	2.80	7.73	0.92
DCCRN	3.70	2.68	3.88	3.18	3.27	-	0.94
Conv-TasNet	5.1	2.84	2.33	2.62	2.51	-	-
TDCGAN	5.12	2.87	4.17	3.46	3.53	9.82	0.95
MetricGAN+	-	3.15	4.14	3.16	3.64	-	-
UNIVERSE	-	3.33	-	-	3.82	-	0.95
CDiffuSE	-	2.52	3.72	2.91	3.10	-	-
SE-Conformer	-	3.13	4.45	3.55	3.82	-	0.95
DB-AIAT	2.81	3.31	**4.61**	3.75	3.96	10.79	0.96
DPT-FSNet	0.91	3.33	4.58	3.72	4.00	-	0.96
DBT-Net	2.91	3.30	4.59	3.75	3.92	-	0.96
**CGA-MGAN**	1.14	**3.47**	4.56	**3.86**	**4.06**	**11.09**	**0.96**

* “-” denotes that the result was not provided in the original paper.

**Table 2 entropy-25-00628-t002:** Results of the ablation study.

Method	PESQ	CSIG	CBAK	COVL	SSNR	STOI
**CGA-MGAN**	**3.47**	**4.56**	**3.86**	**4.06**	**11.09**	**0.96**
w/o Conv. Block	3.37	4.50	3.80	3.97	10.97	0.96
Using Conformer	3.33	4.43	3.72	3.91	10.18	0.96
w/o Gating Decoders	3.43	4.52	3.83	4.02	10.99	0.96
Mag-only	3.42	4.54	3.80	4.03	10.73	0.96
w/o Discriminator	3.37	4.54	3.79	4.00	10.83	0.96

## Data Availability

A publicly available dataset (Voice Bank + DEMAND) was analyzed in this study. The Voice Bank + DEMAND dataset (accessed on 17 December 2021) can be found here: https://datashare.ed.ac.uk/handle/10283/2791.

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
