# Peer review of "CGA-MGAN: Metric GAN Based on Convolution-Augmented Gated Attention for Speech Enhancement"

_entropy, 2023, doi:10.3390/e25040628_

Round 1
Reviewer 1 Report
This paper proposes CGA-MGAN for speech enhancement which can better combine the advantages of convolution and self-attention. The key contribution is to build a generator with an encoder-decoder structure, including gating blocks using the decoupling-style phase-aware method, which can avoid the compensation effect between magnitude and phase.
The authors argue that the proposed structure has lower computational complexity and achieve better results. To verify that, the designed experiments should also compare the computational complexity, e.g. RTF, and further compare the performance with other SOTA models, such as Conv-TasNet and CDiffuSE (diffusion probabilistic model). It would be better to check the generalization capabilities for different databases.
Other comments:
Line 116: the full name of GLU should be given at first.
Line 220: "where" needs to be added on the left.
The location of some equations, such as (1), (17) should be adjusted to the center.
Line 133-134, "previous research has yet to involve the field of speech enhancement", is correct?
In Figure 4, the font size of "Feature from Encoder" should be reduced. Same thing exists in Figure 5.
Line 220, what's the meaning of ReLU^{2}?
In the experimental setup, why slice the training sentences into 2 seconds?
In Section 5, some descriptions of the baselines, such as SE-Conformer, DB-AIAT, DPT-FSNet and DBT-Net, can be moved to Introduction section.
Reviewer 2 Report
This paper presents a deep-learning model based on a "convolution-augmented gated attention MetricGAN" for speech enhancement as an alternative to canonical "conformer" models.
Although the experiment and the model are practical and complex, the paper needs to present the concepts and the components more clearly, especially to justify the model building and the rationale of the approach.
The "Introduction" does not contain a description of the conformer model, which is the experiment baseline. MetricGAN is not described (if not briefly), whereas it is critical for introducing the new model.
Overall, the "Introduction" takes all concepts for granted and is unclear in justifying (either with references or explanations) the reasons for substituting specific conformer components with the newly proposed ones.
Section 2 ("Related Works") should explain MetriGAN with a schema. Moreover, equation terms like D, G, and E are not explained. Overall, this section is too concise.
The description of GAU is too simplistic and does not make the reader appreciate the novelty of the approach. Sentences like "GAU can use SHSA instead of MHSA, which can achieve the same or even better effect as the standard transformer" do not have references associated and are theoretically unjustified.
The "Methodology" misses a detailed comparison with the canonical conformer model, forcing the reader to take the new model for granted without motivation to adopt a particular block sequence instead of a standard conformer one. A comparison with the conformer architecture would have clarified which blocks were substituted and why. Perhaps some explanation is present in the paper, but it is sparse across the sections (from the "Introduction" to the "Results"). Overall, the logic is hard to follow and validate and makes key concepts like CGAU seem arbitrary because they are described but unjustified.
The ablation study tries to add some justification "a posteriori", but it eventually assigns arbitrary interpretations to the gating blocks that are not demonstrated by the results. For example, "the gating block can retain important features in the encoder and suppress irrelevant features, control the information flow of the network and simulate complex interactions", or "the decoupling-style phase-aware method can process coarse-grained regions and fine-grained regions of the spectrum in parallel". Explanations and citations are not reported to demonstrate these conjectures.
Reviewer 3 Report
See the attached annotated file.

Reviewer 4 Report
The manuscript, based on MetricGAN, a generative adversarial network, propose for speech enhancement the CGA-MGAN, which, on top of MetricGAN, incorporate a novel convolution-augmented gated attention mechanism which enables the system to capture local and global correlations in the speech signal.
In the abstract, the "excellent performance" of 3.47 MOS-LQO has to be qualified since in the MOS adjectival scale 5 is excellent, 4 is good and 3 is fair.
Eq. (6) states that the product ^X'm Ym cos Yp equals zero. Is this what you mean?
Eq. (7) states that the product ^X'm Ym sin Yp equals zero. Is this what you mean?
Round 2
Reviewer 1 Report
line 111: "to generator"=>"to generate".
line 320: LGAN should be italic. Other places should also be checked.
Reviewer 2 Report
The modifications are not sufficient to satisfy my previous comments, which asked for a complete restructuring of the paper (and the addition of specific figures I indicated), whereas short modifications were added and the paper structure did not change.
Round 3
Reviewer 2 Report
No response was given to my initially highlighted points. The paper still present the serious faults I highlighted. I cannot understand why this paper is continuosly sent to me for the review.
